# *Neisseria gonorrhoeae*—Susceptibility Trends and Basic Molecular Mapping of Isolates Collected in Israel in 2016–2022

**DOI:** 10.3390/microorganisms13040750

**Published:** 2025-03-26

**Authors:** Zeev Dveyrin, Tal Alon, Andrei Makhon, Israel Nissan, Zohar Mor, Efrat Rorman

**Affiliations:** 1National Public Health Laboratory, Public Health Directorate, Ministry of Health, Tel Aviv 6810416, Israel; tal.alon@phlta.health.gov.il (T.A.); andermachon@gmail.com (A.M.); israeln@moag.gov.il (I.N.); efrat.rorman@moh.gov.il (E.R.); 2Central District Department of Health, Ministry of Health, Ramla 7243003, Israel; zohar.mor@rml.health.gov.il; 3School of Health Sciences, Ashkelon Academic College, Ashkelon 7821100, Israel

**Keywords:** *Neisseria gonorrhoeae* (*NG*), antimicrobial resistance (AMR), sequence types (STs), sexually transmitted infection (STI), surveillance, Israel

## Abstract

*Neisseria gonorrhoeae* (NG) is a globally significant sexually transmitted infection (STI) with increasing antimicrobial resistance (AMR), posing a serious threat to public health. Between 2016 and 2022, the Israeli National NG Reference Center (INNGRC) comprehensively analyzed NG isolates in Israel to determine AMR patterns and sequence types (STs). Antimicrobial susceptibility testing (AST) was performed on 1205 NG isolates using E-test gradient strips, and NG-MAST analysis was conducted on 279 isolates via Sanger sequencing and whole genome sequencing (WGS). Surveillance revealed high resistance rates to ciprofloxacin (54.4%), azithromycin (41.3%), tetracycline, and benzylpenicillin, while all isolates remained susceptible to ceftriaxone and spectinomycin. Multi-drug resistance (MDR) was observed in 8.6% of isolates, and 3% were classified as extensively drug-resistant (XDR). NG-MAST analysis identified 72 distinct STs, with ST292, ST4269, and ST5441 being the most prevalent. ST19665 and ST11461 predominated in 2022, while ST292, ST5441, and ST16169 were more abundant in 2018. The findings highlight the increasing prevalence of AMR in NG in Israel and underscore the importance of continuous surveillance and molecular characterization by reference laboratories like the INNGRC to inform treatment strategies and public health interventions, ultimately reducing the burden of this critical STI.

## 1. Introduction

*Neisseria gonorrhoeae* (NG) is the second-most prevalent bacterial sexually transmitted infection (STI) worldwide. It has developed antimicrobial resistance (AMR) to all previous first-line drugs, such as penicillin, tetracycline, and fluoroquinolones [1,2], leaving the expanded-spectrum cephalosporins ceftriaxone and cefixime as the only antibiotics recommended for the treatment of gonococcal infections [3,4].

The evolution of AMR in NG has emerged as a critical global health challenge over the past decade. This Gram-negative pathogen has demonstrated an exceptional ability to develop resistance to multiple classes of antibiotics, severely limiting treatment options and raising concerns about potentially untreatable gonorrhea infections [5].

One of the most alarming trends has been the increasing resistance to extended-spectrum cephalosporins (ESCs), particularly ceftriaxone, which has long been considered the last line of defense against gonorrhea [6].

In New Zealand, surveillance reports from 2018 to 2022 have tracked the proportion of gonococcal isolates with reduced susceptibility to ceftriaxone [7]. These data provide valuable insights into the evolution of AMR patterns over time and across different demographic groups.

Many nations have observed a steady rise in azithromycin resistance, compromising its effectiveness as part of dual therapy regimens [8].

Ciprofloxacin resistance among NG has remained high globally, with rates exceeding 50% in numerous countries [8].

The emergence of MDR and XDR strains is particularly concerning [9]. These strains often carry multiple resistance determinants, severely limiting treatment options and posing a significant challenge to public health efforts.

N. gonorrhoeae achieves MDR/XDR status through chromosomal mutations and plasmid-mediated resistance. Chromosomal mutations modify antibiotic targets, enhance efflux systems, and alter membrane permeability. Plasmids enable the production of antibiotic-inactivating enzymes and protective proteins. These changes result in four core resistance mechanisms: target modification, enzymatic inactivation, increased efflux, and decreased influx, collectively conferring resistance to multiple antibiotic classes [10].

The COVID-19 pandemic led to decreased testing capacity for NG cultures in many countries, affecting surveillance efforts [11].

There has been notable inter-study variability and frequently small sample sizes in AMR studies, particularly in Africa, making appropriate inter-study and inter-country comparisons difficult [11].

These trends underscore the urgent need for continued global surveillance, the development of new treatment options, and the implementation of effective strategies to combat the growing threat of AMR in NG.

The sequence types (STs) of NG play a significant role in determining the patterns of AMR observed in different regions. The NG ST is defined through molecular analysis based on *por* and *tbpB* gene alleles using the NG multi-antigen sequence typing method (NG-MAST) [9,12].

The most common STs of NG associated with antimicrobial resistance include the following:
1.Globally Prevalent Resistant Sequence Types: ST1901 is frequently associated with decreased susceptibility or resistance to extended-spectrum cephalosporins, particularly ceftriaxone [13,14]. ST1901 has been identified in multiple countries across different continents, indicating its global spread [15].

ST7363 is linked to decreased susceptibility to ceftriaxone and resistance to multiple other antibiotics, and ST-7363 has emerged as a concerning strain, especially in Asian countries [11,16]. ST1407 is associated with multi-drug resistance, including resistance to cephalosporins and azithromycin [12,16].

2.Emerging Resistant Sequence Types: ST13871 is associated with high-level azithromycin resistance, and ST-13871 has been reported in multiple European countries and the United States [16,17]. ST14422 has been identified as a prevalent sequence type in some regions, and ST14422 has been associated with resistance to multiple antibiotics, including tetracycline and ciprofloxacin [13,17]. ST11210 has been linked to decreased susceptibility to extended-spectrum cephalosporins and resistance to fluoroquinolones in some studies [13,16].

The Israeli National NG Reference Center (INNGRC) of the Ministry of Health (MOH) collects all positive NG isolates from all Israeli healthcare organizations (IHOs). It is important to note that Nucleic Acid Amplification Tests (NAATs) for the diagnosis of NG in urine have replaced the traditional methods of microbiological cultures, resulting in fewer NG isolates.

The aim of this study was to describe and analyze the phenotypic AMR patterns and trends of NG isolates collected in Israel, between 2016 and 2022, in conjunction with their NG-MAST results.

Our study describes for the first time the STs and antimicrobial resistance of NG in Israel. Our results may contribute to the understanding of the global spread of specific STs as well as to better treatment and infection containment in Israel.

## 2. Materials and Methods

### 2.1. Study Population and Sample Collection

Clinical isolates of NG were collected by the INNGRC, from all IHOs: health maintenance organizations (HMOs), hospitals’ laboratories (HLs), and STI walk-in clinics (SL). In total, 1205 samples were collected, analyzed, and stored between 2016 and 2022.

### 2.2. Bacterial Growth and Testing

Isolates were cultured on NYC Medium [18] and incubated for 24 h at 35 °C in a 5% CO_2_ atmosphere.

NG isolates were identified by colony morphology, Gram staining, oxidase, catalase, and nitrate tests, and Matrix-Assisted Laser Desorption–Ionization Time-Of-Flight Mass Spectrometry (MALDI-TOF MS, Bruker, Billerica, MA, USA) [19]. All the isolates were stored at −80 °C on Protect Micro-organism Preservation Beads in Cryovial of Technical Service Consultants Ltd. (Heywood, UK) [20]. ATCC NG strain 49226 was used as an internal control for all test methods [21].

### 2.3. Antimicrobial Susceptibility Testing (AST)

A panel of 7 antimicrobials—benzylpenicillin (BEN), ceftriaxone (CTR), cefixime (CIX), azithromycin (AZI), ciprofloxacin (CIP), tetracycline (TET), and spectinomycin (SPE)—was tested by using E-test gradient strips (bioMérieux Marcy-l’E’toile, France) [22]; Minimum Inhibitory Concentrations (MICs) (mg/L) were interpreted according to the European Committee on Antimicrobial Susceptibility Testing clinical breakpoints (EUCAST) [23] for sensitivity (S), intermediate resistance (I), and resistance (R).

NG MDR/XDR susceptibility was defined accordingly:1.Decreased susceptibility to cephalosporins or resistance to azithromycin, along with resistance to at least two other antimicrobials, was defined as MDR-NG;2.Decreased susceptibility to a cephalosporin plus resistance to azithromycin as well as resistance to at least two other antimicrobials was defined as XDR-NG [24].

All isolates were suspended on Mueller Hinton Broth and cultured on GC Medium Base Agar [14].

### 2.4. Molecular Characterization Using NG-MAST Genotyping

Sixty-seven isolates from 2017 were characterized by the NG multi-antigen sequence typing (NG-MAST) system as described [17]. The allele numbers of *por* and *tbpB* and the sequence types (STs) were assigned using a publicly accessible database on the NG-MAST website “https://pubmlst.org/”.

Two hundred and twelve isolates from the years 2018 and 2022 were characterized by whole genome sequencing (WGS). DNA Paired-end libraries were prepared using the Illumina Nextera XT DNA Library Preparation Kit according to Illumina protocols [25]. For sequencing, we utilized the Illumina MiSeq platform using a MiSeq Reagent Kit v2 (500-cycles) (catalog MS-102-2003) or a MiSeq Reagent Kit v3 (600-cycle), (catalog MS-102-3003).

Raw short reads were quality-analyzed with FastQC v0.11.9 [26] and then trimmed with Trimmomatic v0.39 [27] using the parameters “SLIDINGWINDOW: 4:15 MINLEN: 50”. De novo assembly of the short reads was performed using the SPAdes v3.13.1 [28] assembler. The analysis included 1000 ultrafast bootstrap replicates, and the resulting phylogenetic tree was visualized in iTOL. NG-MAST [12] STs were assigned to each isolate using the MLST v2.23.0 [https://github.com/tseemann/mlst accessed on 26 February 2025] tool.

The raw WGS data were submitted to the NCBI SRA for ST confirmation: the Bioproject ID for 2018 is PRJNA1217336 and for 2022 is PRJNA1217803.

## 3. Results

### 3.1. The Demographic Characteristics of the Study Group

A total of 1205 samples were collected and analyzed throughout the study period. All isolates underwent both phenotypic and molecular testing and were identified as NG based on characteristic colony growth on selective agar media, the microscopy of Gram-negative diplococci, biochemical activity tests, and MALDI-TOF MS. The basic demographic information is summarized in Table 1, “Gender distribution”.

Among the 1205 examined samples, 82.9% of the isolates were from males, 9.7% from females, and 7.4% from an unknown gender. The median age of all patients was 33 years for males and 30 years for females across the entire testing period. The gender and age distribution trends for the patients from all IHOs are illustrated in Figure 1, “Gender and age distribution of patients from IHOs”.

This study compares the number of NG-positive isolates collected at the INNGRC with the total number of NG-positive cases reported to the MOH’s Department of Epidemiology (Figure 2).

The number of reported cases was four times higher than the number of NG isolates identified.

### 3.2. Antimicrobial Susceptibility Testing (AST)

A panel of seven antibiotics—cefixime (CIX), azithromycin (AZI), tetracycline (TET), benzylpenicillin (BEN), spectinomycin (SPE), ciprofloxacin (CIP), and ceftriaxone (CTR)—was employed for AST on all NG isolates collected at the INNGRC.

The antimicrobial resistance profile of the 1205 isolates is summarized in Table 2, “NG isolates’ phenotypic antimicrobial resistance profile”.

Notably, 54.4% of the isolates exhibited phenotypic resistance to CIP, 41.3% to AZI, 8.7% to CIX, 27.6% to TET, and 15.9% to BEN, while no resistance was observed for SPE and CTR.

Figure 3, “Trends in antimicrobial resistance of NG tested in Israel: 2016–2022 (%)”, presents the antimicrobial resistance profile for each antibiotic individually.

There was no observed resistance to CTR and SPE, and from 82% up to 99% of the samples were sensitive to CIX. For certain antibiotics such as BEN, AZI, CIP, and TET, three levels of susceptibility were noted: resistant (R), sensitive (S), and intermediate resistance (I). The intermediate resistance level was predominant for BEN (60–90%) and TET (40–60%), while fluctuations in R/S/I levels for other drugs did not show clear trends. An analysis of the combined effect of various antibiotics indicated that 104 isolates were classified as NG-MDR (8.6%) and 36 isolates as NG-XDR (3%). The distribution of NG MDR/XDR isolates is illustrated in Figure 4, “NG MDR/XDR distribution in Israel: 2016–2022”, showing a low level of NG-MDR isolates (5%) with no NG-XDR detected in the last two years of the study (2021–2022).

### 3.3. Molecular Characterization by NG-MAST Genotyping

Seventy-two different NG-MAST sequence types were identified among the 279 sequenced isolates collected during the years 2017–2018 and 2022. Only for 2017 were the NG-MAST results for all samples identified according to the numbers of the two alleles (*por* and *tbpB*), and for 2018 and 2022, NG-MAST results were obtained from WGS data. The full WGS data of NG isolates in Israel will appear in a separate article. Of these, 36 were represented by a single isolate, 11 by double isolates, and 4 by triple isolates. Twenty-one clusters contained more than four NG isolates. Notably, eight (11.3%) of the STs had not been previously reported in other countries. The interruption of NG-MAST characterization from 2019 to 2021 was due to all molecular testing resources being redirected to COVID-19 diagnosis and characterization.

All sequence type clusters comprising at least four identical STs are presented in Table 3, “Distribution of NG-MAST sequence types in *Neisseria gonorrhoeae* isolates in Israel: 2017–2018 and 2022”.

Twelve of the most common STs are shown in Figure 5, “Maximum likelihood phylogenetic tree of Neisseria gonorrhoeae isolates collected in 2018 and 2022 in Israel”.

This phylogenetic tree was constructed to visualize the relationships among isolates. The most abundant STs were ST19665 (31), ST11461 (30), ST4269 (16), ST292 (16), ST14994 (14), ST5441 (13), ST19762 (10), and ST2318 (9), accounting for a significant proportion of the typed isolates. Notably, the distribution of STs varied among the sampled years. ST19665 and ST11461 predominated in 2022, whereas ST292, ST5441, and ST16169 were more abundant in 2018. While some STs formed tight clusters on the tree, indicating close phylogenetic relationships, other branches and ST distributions demonstrated substantial genetic diversity.

The most common NG-MAST types and their corresponding antimicrobial resistance results are summarized in Table 4.

The most frequently observed STs in Israel included the following: ST19665 (*n =* 31), ST11461 (*n* = 25), ST292 (*n* = 16), ST4269 (*n* = 16), ST5441 (*n* = 13), ST14994 (*n* = 10), and ST2318 (*n* = 9).

## 4. Discussion

Our study provides the first comprehensive overview of antimicrobial resistance (AMR) profiles in Neisseria gonorrhoeae (NG) isolates collected over a seven-year period (2016–2022) in Israel, along with the inaugural genetic characterization using NG-MAST methodology. While previous research examined various aspects of STIs in Israel, including gonococcal antibiotic sensitivity [29,30,31], one notable study [32] analyzed antibiotic susceptibility patterns from two major healthcare providers (HMOs and SL). Our current research expands upon these findings through a more comprehensive temporal analysis and genetic characterization, enhancing our understanding of NG epidemiology and evolution in Israel.

### 4.1. Demographic Patterns and Surveillance

The NG AMR monitoring system revealed distinctive demographic patterns. Male samples constituted 84.2% across all Israeli health organizations (IHOs), yielding a male-to-female ratio of 12.3:1. This disparity likely reflects the prevalence of asymptomatic infections in females [33,34]. Female representation in Israel (8.4%) was notably lower than in European data (15.5%), with a higher median age (33 versus 25 years) compared to European surveillance (2009–2017) [2]. This age difference may reflect Israel’s universal military conscription and post-service travel patterns.

Age distribution analysis showed that 76% of isolates came from individuals aged 21–40, with 15% from those aged 41–60. Previous research suggests potential engagement with sex workers among these age groups [35].

### 4.2. Surveillance Systems and Reported Trends

As a notifiable disease in Israel [29], reported gonorrhea cases exceeded INNGRC isolates fourfold (Figure 2), primarily due to the widespread adoption of molecular diagnostic methods [36]. While these methods offer rapid, reliable results without bacterial culture, live cultures remain essential for phenotypic AST, cluster identification, and genomic analyses [37].

The World Health Organization’s target of reducing gonorrhea incidence by 90% [38]—adopted by Israel’s national STI program for 2025 [26]—faces significant challenges. Our data show no substantial decrease in reported cases between 2016 and 2022, mirroring European trends [39]. Contributing factors include the following:1.International travel-mediated pathogen importation;2.Evolving sexual behaviors;3.Reduced condom use, potentially linked to increased PrEP adoption;4.Enhanced anonymous testing availability;5.Establishment of dedicated sexual health clinics within HMOs.

### 4.3. Non-Endemic Status and AMR Implications

Israel’s non-endemic status significantly influences AMR trend interpretation. The observed patterns likely reflect imported strains and limited local transmission, supported by the following:1.Travel-associated gonorrhea prevalence [10,40];2.High genetic diversity in NG-MAST results [41];3.Variable resistance profiles consistent with imported strain patterns [42].

Despite these considerations, our AMR findings align with global research [43,44,45], suggesting Israel’s integration into the global NG transmission network. Similarly to European monitoring [46], we found no resistance to ceftriaxone (CTR) or spectinomycin (SPE). Current guidelines recommend single-dose injectable CTR for NG treatment in Israel [47].

### 4.4. Antimicrobial Resistance Patterns

Ciprofloxacin resistance has increased significantly to 54.4% ± 13.4%—more than double the 2002–2007 rate of 26.1%—aligning with global trends [48,49,50,51]. The monitoring of intermediate (I) resistance alongside resistant (R) and susceptible (S) levels proved crucial, as demonstrated by benzylpenicillin patterns showing increased intermediate resistance (62% to 86%) despite decreased full resistance. Similar patterns were documented in Northern Spain [52], where BEN resistance rates were R = 1.3%, I = 81.7%, and S = 17.0%, with tetracycline also exhibiting high intermediate resistance (I = 43.1%).

To combat antibiotic resistance, personalized antibiotic therapies show promise, incorporating AI-driven prediction models and rapid diagnostics for tailored treatments based on individual profiles and local resistance patterns [53]. Prompt, appropriate antibiotic therapy with the minimal recommended duration can reduce MDR strain emergence [54].

### 4.5. Multi-Drug Resistance Trends

We observed a decreasing trend in XDR and MDR levels from 2016 to 2022. MDR-NG rates decreased from 23.4% to approximately 4% (2021–2022), while XDR-NG cases (9.9% in 2016) have been absent in recent years, comparable to Canadian findings [21,55]. Global resistance to ceftriaxone and azithromycin has been documented in the United Kingdom, Australia [56], and France, particularly in patients with a travel history to Cambodia [57].

### 4.6. Genetic Diversity and Sequence Types

Our analysis revealed a high degree of genetic diversity among N. gonorrhoeae isolates, as illustrated in the phylogenetic tree (Figure 5). A clear temporal shift was observed in the dominant sequence types, with ST292 and ST5441 prevalent in 2018, while ST19665 and ST11461 predominated in 2022, alongside the significant presence of ST14994 and ST19762. This shift suggests either rapid clonal expansion or the successful establishment of new strains.

Between 2016 and 2022, we identified 27 NG-MAST sequence types circulating in Israel, reflecting a diverse genetic landscape. These types were also previously documented in European surveys from 2009 to 2010 and 2013. Several prevalent Israeli sequence types (ST4269, ST5441, ST2318, ST2997, ST11547, ST16169) were under-represented in recent Euro-GASP surveys [2,46]. In contrast, six of the most prevalent European sequence types (ST21, ST292, ST2992, ST5624, ST5793) were detected in Israel [58,59].

Notably, ST4269, classified as XDR/MDR, was absent in 2022, while other XDR/MDR-NG clusters (such as ST4269 and ST2318) were identified in 2017–2018 and 2017–2022. Antimicrobial resistance (AMR) diversity within these clusters varied significantly. For instance, ST1993 exhibited full sensitivity, while ST19972 demonstrated 50% resistance to azithromycin.

Analyzing the correlations between genotypes, clinical presentations, and international comparisons not only enhances our understanding of NG epidemiology in Israel but also underscores the critical importance of maintaining molecular surveillance to detect emerging resistant strains, potentially imported through international travel.

At this stage, we were unable to collect detailed clinical data. However, previous research has linked specific sequence types to distinct clinical presentations and antimicrobial resistance profiles. For example, ST1901 is frequently associated with decreased susceptibility to ceftriaxone, while ST7363 has been linked to decreased susceptibility to ceftriaxone and resistance to multiple other antibiotics. Although ST1901 and ST7363 were not identified in our collection, the high rates of resistance to ciprofloxacin (54.4%) and azithromycin (41.3%) observed in our isolates suggest the presence of other resistant sequence types.

Further studies correlating the identified sequence types (e.g., ST292, ST4269, ST5441, ST19665, and ST11461) with patient demographics, clinical outcomes, and detailed resistance profiles are needed to fully understand the clinical implications of these circulating strains in Israel. Such research would provide valuable insights for the development of targeted treatment strategies and inform public health interventions.

### 4.7. Future Directions and Recommendations

The WHO recommends ongoing AMR surveillance as an early warning mechanism for community resistance. Israel currently lacks a national AMR NG monitoring strategy, unlike established programs such as WHO’s GASP, Euro-GASP, and others [50,60,61]. Implementing an annual GASP/GISP-like program could be crucial for monitoring Israel’s STI landscape and for the early detection of resistant strains.

## 5. Conclusions

This study reveals significant NG antimicrobial resistance rates, with high resistance to ciprofloxacin (54.4%), azithromycin (41.3%), and tetracycline (27.6%) while maintaining susceptibility to ceftriaxone and spectinomycin. Among the 72 distinct NG-MAST sequence types identified, ST19665, ST11461, ST292, and ST4269 predominated. While 8.6% of isolates showed MDR and 3% exhibited XDR, resistant strain prevalence decreased over the study period.

## 6. Study Limitations and Strengths

This study faces several constraints. The sample predominantly consists of isolates from men, limiting gender representation. Geographic coverage is incomplete, and the source of all isolates cannot be definitively confirmed as IHOs. The inability to link behavioral data with laboratory findings restricts the interpretation of results. The shift toward molecular diagnostics has reduced the availability of cultured isolates, and only a portion underwent NG-MAST analysis, potentially limiting the genetic diversity of the data.

Despite these limitations, this study exhibits notable strengths. It analyzes 1205 Neisseria gonorrhoeae isolates over seven years (2016–2022) from three distinct IHOs. The research employs a multifaceted approach, combining phenotypic antimicrobial susceptibility testing with NG-MAST genotyping. This robust methodology, coupled with the study’s extensive dataset and diverse sampling, provides current and relevant antimicrobial resistance data. Consequently, the findings offer valuable insights for updating treatment guidelines, making a significant contribution to N. gonorrhoeae epidemiology and resistance surveillance in Israel.

## Figures and Tables

**Figure 1 microorganisms-13-00750-f001:**
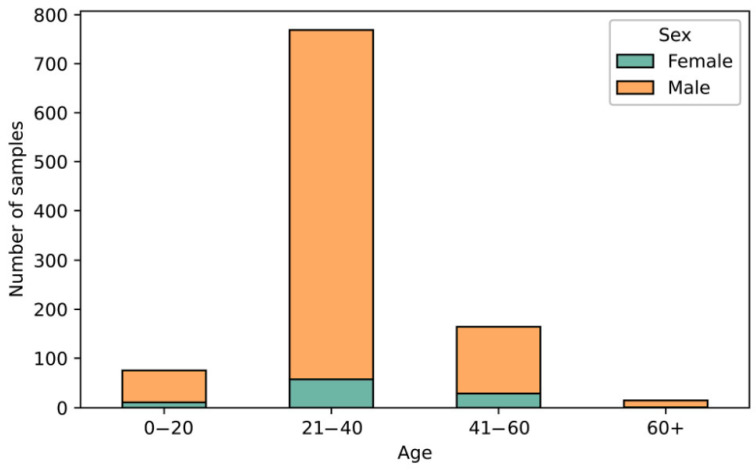
Gender and age distribution of patients from IHOs.

**Figure 2 microorganisms-13-00750-f002:**
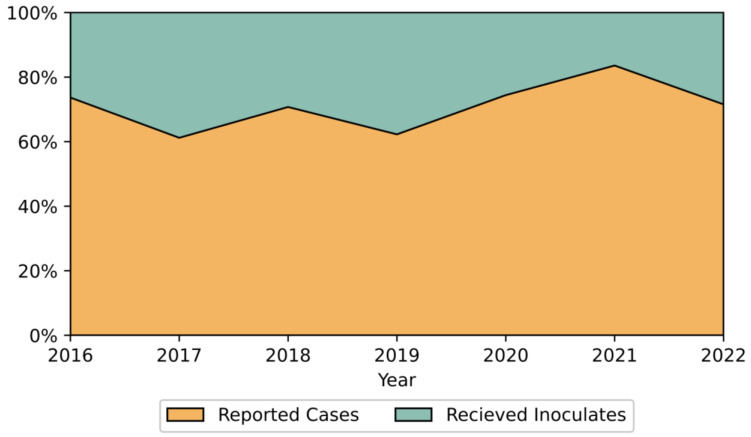
Comparison of reported cases and received positive isolates.

**Figure 3 microorganisms-13-00750-f003:**
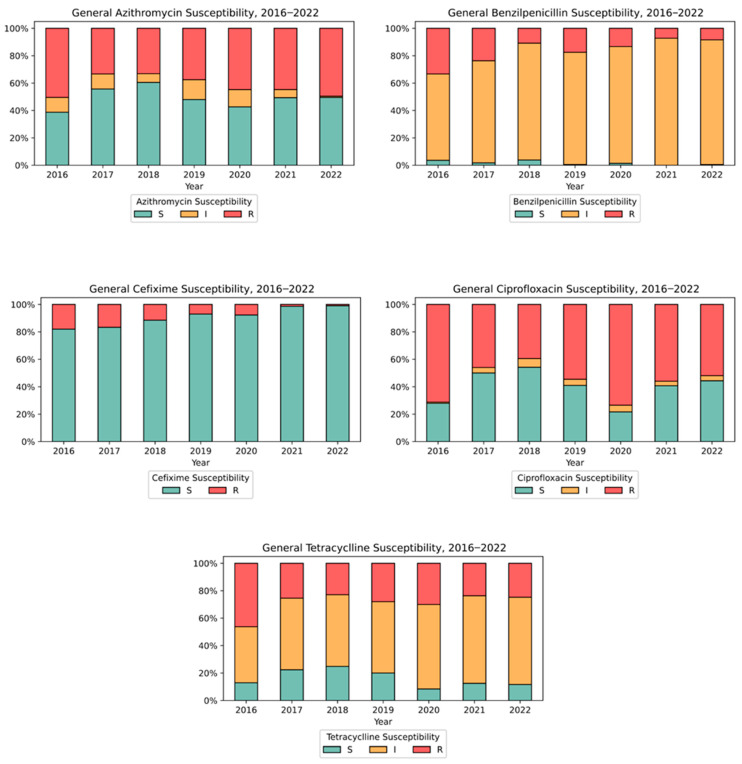
Trends in antimicrobial resistance of NG tested in Israel: 2016–2022 (%).

**Figure 4 microorganisms-13-00750-f004:**
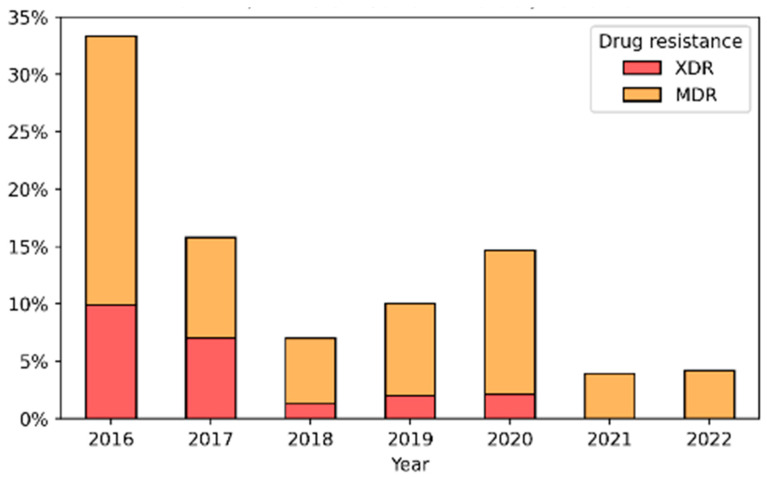
NG MDR/XDR distribution in Israel: 2016–2022.

**Figure 5 microorganisms-13-00750-f005:**
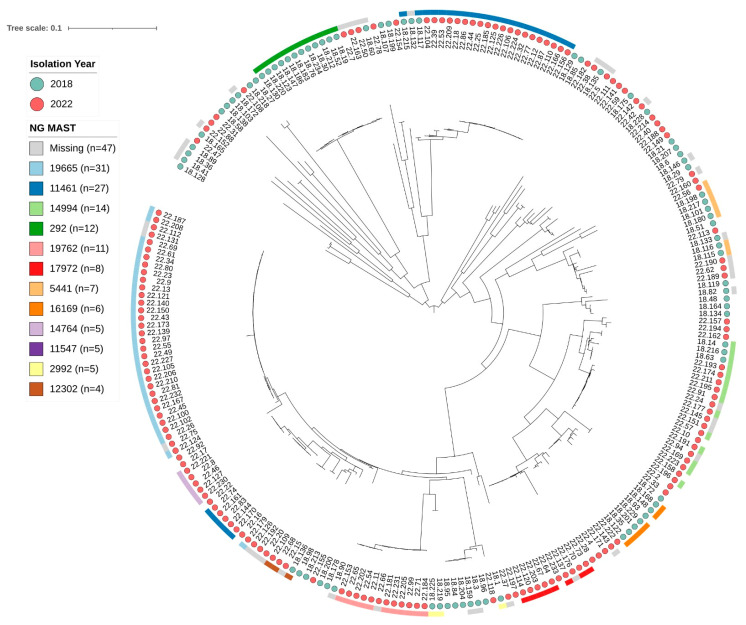
Maximum likelihood phylogenetic tree of Neisseria gonorrhoeae isolates collected in 2018 and 2022 in Israel.

**Table 1 microorganisms-13-00750-t001:** Gender distribution.

Year	Number of Isolates	Ratio—Male to Female	Median Age—Female/Male
2016	111	14.7	27/29
2017	228	6.8	31/30
2018	157	5.8	31/29
2019	200	14.6	41/31
2020	143	16.1	46/29
2021	152	8.9	33/28.5
2022	214	15.3	30/32
All years	1205	10 *	33 */30 *
		12.3 **	32/30 **

*—average result. **—median result.

**Table 2 microorganisms-13-00750-t002:** NG isolates’ phenotypic antimicrobial resistance profile.

Year	Number of Isolates	Resistance *						
		CIX	AZI	TET	BEN	SPE	CIP	CTR
2016	111	20 (18.0%)	56 (50.5%)	50 (46.3%)	37 (33.3%)	0	79 (71.2%)	0
2017	228	38 (16.7%)	76 (33.3%)	58 (25.4%)	54 (23.7%)	0	105 (46.1%)	0
2018	157	18 (11.5%)	52 (33.1%)	36 (22.9%)	17 (10.8%)	0	62 (39.5%)	0
2019	200	14 (7.0%)	75 (37.5%)	56 (28.0%)	35 (17.5%)	0	109 (54.5%)	0
2020	143	11 (7.7%)	64 (44.8%)	43 (30.1%)	19 (13.3%)	0	105 (73.4%)	0
2021	152	2 (1.3%)	68 (44.7%)	36 (23.7%)	11 (7.2%)	0	85 (55.9%)	0
2022	214	2 (0.9%)	106 (49.5%)	53 (24.8%)	18 (8.4%)	0	111 (51.9%)	0
Total (Ave% ± SD)	1205	105 (8.7% ± 6.3)	497 (41.2% ± 7.1)	332 (27.6% ± 8.7)	191 (15.9% ± 9.6)	0	656 (54.4% ± 13.4)	0

Abbreviations: cefixime = CIX; azithromycin = AZI; tetracycline = TET; benzylpenicillin = BEN; spectinomycin = SPE; ciprofloxacin = CIP; ceftriaxone = CTR. The data are the number of isolates, with % where relevant. * Resistance breakpoints determined by EUCAST.

**Table 3 microorganisms-13-00750-t003:** Distribution of NG-MAST sequence types in Neisseria gonorrhoeae isolates in Israel: 2017–2018 and 2022.

Year	Number of Isolates	Number of NG-MAST STs	Most Common NG-MAST STs (Number of Isolates) *
2017	78	72	ST4269 (13), ST2318 (7), ST2997 (7), ST5441 (6), ST5049 (5), ST292 (4), ST5624 (4)
2018	77	72	ST292 (12), ST5441 (6), ST16169 (6), ST2992 (5), ST11547 (5), ST11461 (4)
2022	124	72	ST19665 (31), ST11461 (25), 14994 (10), 19762 (10), 17972 (8), 14764 (5)
Total	279	72	ST19665 (31), ST11461 (30), ST4269 (16), ST292 (16), ST14994 (14), ST5441 (13), ST19762 (10), ST2318 (9), ST17972 (8), ST2992 (7), ST11547 (7), ST2997 (7), ST16169 (6), ST5049 (5), ST9208 (5), ST14764 (5), ST3935 (4), ST14760 (4), ST14051 (4), ST5624 (4), ST12302 (4)

* Represented by at least 4 isolates.

**Table 4 microorganisms-13-00750-t004:** Distribution of most prevalent STs’ antimicrobial patterns.

NG-MAST	*n* *	MDR,%	XDR,%	CIP **	BEN **	TET **	AZI **	CIX **
ST19665	31	0	0	S(96.8%) R(3.2%)	I	S(3.2%) I(93.5%) R(3.2%)	S(9.7%) R(90.3%)	S(96.8%) R(3.2%)
ST11461	30	0	0	S(83.3%) R(16.7%)	I	S(3.3%) I(13.3%) R(83.3%)	S(86.7%) R(13.3%)	S
ST292	16	0	0	S	I	S(25.0%) I(75.0%)	S(87.5%) I(6.2%) R(6.2%)	S
ST4269	16	29.4	64.7	R	I(17.6%) R(82.4%)	I(5.9%) R(94.1%)	I(5.9%) R(94.1%)	S(29.4%) R(70.6%)
ST14994	14	0	0	R	I(92.9%) R(7.1%)	S(50.0%) I(50.0%)	S(85.7%) I(7.1%) R(7.1%)	S
ST5441	13	0	0	S	S(7.7%) I(92.3%)	S	S(30.8%) I(30.8%) R(38.5%)	S
ST19762	10	10	0	R	I	S(10.0%) I(80.0%) R(10.0%)	S(20.0%) R(80.0%)	S
ST2318	9	45.5	9.1	S(9.1%) R(90.9%)	I(81.8%) R(18.2%)	I(18.2%) R(81.8%)	S(27.3%) R(72.7%)	S(90.9%) R(9.1%)
ST17972	8	12.5	0	R	I	I(87.5%) R(12.5%)	S(25.0%) R(75.0%)	S
ST2997	7	0	0	S	I	I(71.4%) R(28.6%)	I(14.3%) R(85.7%)	S
ST2992	7	0	0	S(85.7%) I(14.3%)	I	S(14.3%) I(85.7%)	I(28.6%) R(71.4%)	S
ST11547	7	14.3	0	R	I(85.7%) R(14.3%)	S(28.6%) I(71.4%)	S(71.4%) I(14.3%) R(14.3%)	S(14.3%) R(85.7%)
ST16169	6	0	0	R	I	S(33.3%) I(66.7%)	S(66.7%) R(33.3%)	S
ST9208	5	0	0	S	I	S(40.0%) I(60.0%)	S(40.0%) I(20.0%) R(40.0%)	S
ST5049	5	0	0	S	I	I(80.0%) R(20.0%)	R	S
ST14764	5	0	0	R	I	I	R	S
ST14760	4	0	0	S	S(25.0%) I(50.0%) R(25.0%)	S(50.0%) I(50.0%)	S	S
ST12302	4	50	0	I(25.0%) R(75.0%)	I	I(50.0%) R(50.0%)	R	S
ST14051	4	50	0	R	I	I(25.0%) R(75.0%)	S(50.0%) R(50.0%)	S
ST5624	4	50	0	R	I(50.0%) R(50.0%)	I	R	S
ST3935	4	0	0	S	I	I	R	S

* Represented by *n =* 4 isolates. ** Abbreviations: ciprofloxacin = CIP; tetracycline = TET; azithromycin = AZI; benzylpenicillin = BEN; cefixime = CIX.

## Data Availability

The original data presented in this study are openly available at NCBI SRA Bioproject 1 at: https://www.ncbi.nlm.nih.gov/sra/?term=PRJNA1217336 (accessed on 26 February 2025) and at https://www.ncbi.nlm.nih.gov/sra/?term=PRJNA1217803 (accessed on 26 February 2025).

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
