# Peer review of "Neisseria gonorrhoeae*—Susceptibility Trends and Basic Molecular Mapping of Isolates Collected in Israel in 2016–2022"

_microorganisms, 2025, doi:10.3390/microorganisms13040750_

Round 1
Reviewer 1 Report (Previous Reviewer 2)
Comments and Suggestions for Authors
I thank the authors for their new submission. The authors have incorporated all the information requested by the other reviewers into the main text. The manuscript has significantly improved compared to the previous version, and I believe it can be accepted in its current form, pending the Editor’s final decision.
Author Response
Comment 1: I thank the authors for their new submission. The authors have incorporated all the information requested by the other reviewers into the main text. The manuscript has significantly improved compared to the previous version, and I believe it can be accepted in its current form, pending the Editor’s final decision.
Respond 1: Thank you for your comment.
Reviewer 2 Report (New Reviewer)
Comments and Suggestions for Authors
In general, the work is well written and understandable. It also makes a significant contribution to determining the genotypes of circulating NG. I would therefore suggest some minor corrections to the manuscript:
Line 52 - specify what these resistance determinants would be, including the resistance mechanisms that NG realises in order to be MDR and XDR.
Figure 3 is very interesting for demonstrating in a didactic way, especially the tendency of intermediate isolates.
In the discussion, clinical data obtained from the literature could be correlated with the genotypes identified. This would give greater depth to the discussion.
As well as comparing the isolates obtained with reports from other countries, which would contribute to the hypothesis of imported cases.
Comments on the Quality of English LanguageThe language is adequate.
Author Response
Comment 1: Line 52 - specify what these resistance determinants would be, including the resistance mechanisms that NG realises in order to be MDR and XDR.
Respond 1: Thank you for pointing this out. The following paragraph was added to the manuscript as a Line 54-59.
"N. gonorrhoeae achieves MDR/XDR status through chromosomal mutations and plasmid-mediated resistance. Chromosomal mutations modify antibiotic targets, enhance efflux systems, and alter membrane permeability. Plasmids enable production of antibiotic-inactivating enzymes and protective proteins. These changes result in four core resistance mechanisms: target modification, enzymatic inactivation, increased efflux, and decreased influx, collectively conferring resistance to multiple antibiotic classes [8]."
Comment 2: Figure 3 is very interesting for demonstrating in a didactic way, especially the tendency of intermediate isolates.
Respond 2: Thank you for your comment.
Comment 3: In the discussion, clinical data obtained from the literature could be correlated with the genotypes identified. This would give greater depth to the discussion. As well as comparing the isolates obtained with reports from other countries, which would contribute to the hypothesis of imported cases.
Respond 3: Thank you for pointing this out. The following paragraph "4.6 Genetic Diversity and Sequence Types" Line 321-354 has been revised to address your comments:
"Our analysis revealed a high degree of genetic diversity among N. gonorrhoeae isolates, as illustrated in the phylogenetic tree (Figure 5). A clear temporal shift was observed in the dominant sequence types, with ST292 and ST5441 prevalent in 2018, while ST19665 and ST11461 predominated in 2022, alongside a significant presence of ST14994 and ST19762. This shift suggests either rapid clonal expansion or the successful establishment of new strains.
Between 2016 and 2022, we identified 27 NG-MAST sequence types circulating in Israel, reflecting a diverse genetic landscape. These types were also previously documented in European surveys from 2009-2010 and 2013. Several prevalent Israeli sequence types (ST4269, ST5441, ST2318, ST2997, ST11547, ST16169) were underrepresented in recent Euro-GASP surveys [2, 47]. In contrast, six of the most prevalent European sequence types (ST21, ST292, ST2992, ST5624, ST5793) were detected in Israel [59, 53].
Notably, ST4269, classified as XDR/MDR, was absent in 2022, while other XDR/MDR-NG clusters (such as ST4269 and ST2318) were identified in 2017-2018 and 2017-2022. Antimicrobial resistance (AMR) diversity within these clusters varied significantly. For instance, ST1993 exhibited full sensitivity, while ST19972 demonstrated 50% resistance to azithromycin.
Analyzing the correlations between genotypes, clinical presentations, and international comparisons not only enhances our understanding of NG epidemiology in Israel but also underscores the critical importance of maintaining molecular surveillance to detect emerging resistant strains, potentially imported through international travel.
At this stage, we were unable to collect detailed clinical data. However, previous research has linked specific sequence types to distinct clinical presentations and antimicrobial resistance profiles. For example, ST1901 is frequently associated with decreased susceptibility to ceftriaxone, while ST7363 has been linked to decreased susceptibility to ceftriaxone and resistance to multiple other antibiotics. Although ST1901 and ST7363 were not identified in our collection, the high rates of resistance to ciprofloxacin (54.4%) and azithromycin (41.3%) observed in our isolates suggest the presence of other resistant sequence types.
Further studies correlating the identified sequence types (e.g., ST292, ST4269, ST5441, ST19665, and ST11461) with patient demographics, clinical outcomes, and detailed resistance profiles are needed to fully understand the clinical implications of these circulating strains in Israel. Such research would provide valuable insights for the development of targeted treatment strategies and inform public health interventions".
This manuscript is a resubmission of an earlier submission. The following is a list of the peer review reports and author responses from that submission.
Round 1
Reviewer 1 Report
Comments and Suggestions for Authors
Comments
I have reviewed the article title "Neisseria gonorrhoeae - Susceptibility Trends and Basic 2 Molecular Mapping of the Isolates Collected in Israel 3 During 2016–2022" and I have found some major flaws, which need extensive revision before further processing.
Lines 150-152. These lines can be correct as “Comparing the reported cases and received positive inoculates: The study compares the number of NG positive inoculates collected at the INNGRC with the total number of NG positive cases reported to the MOH's Department of Epidemiology (Figure 2).”
Table 2. the first column heading should be “s.no” or “serial number”…
Why these seven antibiotics—Cefixime, Azithromycin, Tetracycline, Benzylpenicillin, Spectinomycin, Ciprofloxacin, and Ceftriaxone selected for MIC testing? Are there any other 3rd generation antibiotics used for antimicrobial susceptibility testing?
What was the authors criteria for claiming the MDR, and XDR strains?
Most importantly, the authors did not report WGS data for any strain, why?
Have these bacteria isolated by the authors? Are the raw data submitted the NCBI SRA database for evaluation?
These questions should be answered one by one.
Author Response
Comment 1: Lines 150-152. These lines can be correct as “Comparing the reported cases and received positive inoculates: The study compares the number of NG positive inoculates collected at the INNGRC with the total number of NG positive cases reported to the MOH's Department of Epidemiology (Figure 2).”
Respond 1: Thank you for pointing this out. We agree with this comment and will update our manuscript accordingly. The lines 150-152 were corrected
Comment 2: Table 2. the first column heading should be “s.no” or “serial number”…
Respond 2: Thank you for pointing this out. We agree with this comment. The first column of the Table 2 is not necessary and was deleted. Our manuscript was updated accordingly.
Comment 3: Why these seven antibiotics—Cefixime, Azithromycin, Tetracycline, Benzylpenicillin, Spectinomycin, Ciprofloxacin, and Ceftriaxone selected for MIC testing?
Respond 3: According to the physician's requests, the INNGRC receives positive NG inoculates for confirmation and susceptibility testing with 7 antibiotics. These inoculates were preliminarily examined in primary laboratories of health maintenance organizations and hospitals with the same drug panel.
Comment 4: Are there any other 3rd generation antibiotics used for antimicrobial susceptibility testing?
Respond 4: Only Cefixime and Ceftriaxone were used for antimicrobial susceptibility testing as 3rd generation antibiotics
Comment 5: What was the authors criteria for claiming the MDR, and XDR strains?
Respond 5: For MDR/XDR claiming we used the criteria described in Lines 111-115
Comment 6: Most importantly, the authors did not report WGS data for any strain, why?
Respond 6: In our study, we focused only on ST (Sequence Type), which was obtained using the NG-MAST method in 2017, while in 2018 and 2022, all isolates were sequenced by the WGS method, with only using the ST data. We plan to report the results of the NG WGS data study in our next article.
Comment 7: Have these bacteria been isolated by the authors?
Respond 7: No, all the positive NG isolates have been sent to the INNGRC from primary medical labs for results confirmation, AST, ST definition, and NG strain bank maintenance.
Comment 8: Are the raw data submitted to the NCBI SRA database for evaluation?
Respond 8: No, but the sequence types (STs) were assigned using the publicly accessible database on the NG-MAST website (available at: https://pubmlst.org/).
Reviewer 2 Report
Comments and Suggestions for Authors
The authors conducted an intriguing study aiming to describe and analyze the phenotypic antimicrobial resistance patterns and trends of Neisseria gonorrhoeae isolates collected in Israel between 2016 and 2022. This manuscript is highly relevant given the prevalence of Neisseria gonorrhoeae infections and the increasingly significant emergence of multidrug-resistant strains, which render this pathogen challenging to treat. Below are my comments:
- Line 48: Are the authors specifically referring to Neisseria gonorrhoeae or to the global resistance of bacterial strains to ciprofloxacin? Unfortunately, both conditions are on the rise. This concept could be further elaborated upon in the text.
- Materials and Methods, Line 95: Are data on the age of patients from whom the bacterial isolates were obtained available? Providing information such as the mean age with standard deviation or the age range (minimum-maximum) would be valuable. This is important due to the wide age differences associated with varying prevalence rates of multidrug-resistant strains, as shown in Figure 1.
- The results are clear and appropriate for the manuscript's purpose.
- Line 233: Do male patients exhibit a higher rate of resistance? Could this be attributed to symptomatic infections? The authors should provide their perspective on this point.
- Line 303: According to the current guidelines in Israel, what are the first-line antibiotics recommended for Neisseria gonorrhoeae infections?
- In the discussion, the authors should provide a more in-depth analysis of strategies aimed at reducing the emergence of multidrug-resistant bacterial strains. One potential solution is the use of personalized antibiotic therapies, characterized by shorter treatment durations tailored to the patient's characteristics and the type of infection. This approach has been robustly demonstrated in pediatric studies (see, for example, 10.3390/children9111647 for urinary tract infections). I recommend expanding on this concept in the main text.
- Line 373: Include the manuscript’s strengths as it is a high-quality study.
- If possible, add a brief conclusion summarizing the authors' findings.
Minor improvements in the English translation are necessary.
Author Response
Comment 1: Line 48: Are the authors specifically referring to Neisseria gonorrhoeae or to the global resistance of bacterial strains to ciprofloxacin? Unfortunately, both conditions are on the rise. This concept could be further elaborated upon in the text.
Respond 1: Thank you for pointing this out. Indeed, it was not clear. We change the sentence (Line 48) from: "Ciprofloxacin resistance remained high globally, with rates exceeding 50% in numerous countries" to "Ciprofloxacin resistance among NG remained high globally, with rates exceeding 50% in numerous countries".
Comment 2: Materials and Methods, Line 95: Are data on the age of patients from whom the bacterial isolates were obtained available? Providing information such as the mean age with standard deviation or the age range (minimum-maximum) would be valuable. This is important due to the wide age differences associated with varying prevalence rates of multidrug-resistant strains, as shown in Figure 1.
Respond 2: Thank you for pointing this out. We have the raw data on the patient's age. The age range of the patient according to gender is provided in Figure 1 and the median age is provided in Table 1 and the Lines 241-244.
Comment 3: The results are clear and appropriate for the manuscript's purpose.
Respond 3: Thank you very much for your comment.
Comment 4: Line 233: Do male patients exhibit a higher rate of resistance? Could this be attributed to symptomatic infections? The authors should provide their perspective on this point.
Respond 4: Thank you for pointing this out. The male patients' inoculates do not exhibit a higher rate of resistance. Our retrospective data do not allow us to draw statistically significant conclusions about any correlation between gender and antibiotic resistance.
Comment 5: Line 303: According to the current guidelines in Israel, what are the first-line antibiotics recommended for Neisseria gonorrhoeae infections?
Respond 5: Thank you for pointing this out. The first-line antibiotic recommended for Neisseria gonorrhoeae infections in Israel is ceftriaxone. It's mentioned now in Lines 301-302: "According to current medical guidelines recommendation, a single dose of injectable CTR utilized in Israel for the treatment of NG [62]."
Comment 6: In the discussion, the authors should provide a more in-depth analysis of strategies aimed at reducing the emergence of multidrug-resistant bacterial strains. One potential solution is the use of personalized antibiotic therapies, characterized by shorter treatment durations tailored to the patient's characteristics and the type of infection. This approach has been robustly demonstrated in pediatric studies (see, for example, 10.3390/children9111647 for urinary tract infections). I recommend expanding on this concept in the main text.
Respond 6: Thank you for pointing this out. The following paragraph was added to the manuscript as a Line 322-330:
"To prevent antibiotic resistance, multiple strategies can be employed. One promising approach is personalized antibiotic therapies: prescribing targeted antibiotics promptly, following local guidelines, and optimizing treatment duration. Recent advancements like AI-driven prediction models and rapid diagnostics enable tailored treatments based on individual patient profiles and local resistance patterns [64]. Physicians can reduce the emergence of NG MDR strains by using the most appropriate antibiotic therapy quickly after diagnosis and for the minimal recommended duration [63]. While an effective strategy, this approach should be integrated into a comprehensive antibiotic management framework."
Comment 7: Line 373: Include the manuscript’s strengths as it is a high-quality study.
Respond 7: Thank you for pointing this out. The following paragraph was added to the manuscript as a Line 392-398.
"At the same time, this comprehensive study exhibits notable strengths, analyzing 1,205 Neisseria gonorrhoeae isolates over seven years (2016-2022) from three distinct Integrated Health Organizations. The research employs a multifaceted approach, combining phenotypic antimicrobial susceptibility testing with NG-MAST genotyping. This robust methodology, coupled with the study's extensive dataset and diverse sampling, provides current and relevant antimicrobial resistance data. Consequently, the findings offer valuable insights for updating treatment guidelines, making a significant contribution to N. gonorrhoeae epidemiology and resistance surveillance".
Comment 8: If possible, add a brief conclusion summarizing the authors' findings.
Respond 8: Thank you for pointing this out. The following paragraph was added to the manuscript as a Line 372-378.
"5. Conclusion
This study reveals significant antimicrobial resistance rates in Neisseria gonorrhoeae, with high resistance to ciprofloxacin (54.4%), azithromycin (41.3%), and tetracycline (27.6%). However, the bacteria remain fully susceptible to ceftriaxone and spectinomycin. The research identified 72 distinct NG-MAST sequence types, with ST19665, ST11461, ST292, and ST4269 being the most prevalent. Notably, 8.6% of isolates were classified as multi-drug resistant (MDR), while 3% were extensively drug-resistant (XDR). Encouragingly, a declining trend in resistant strains was observed over the study period."
Comment 9: Minor improvements in the English translation are necessary.
Respond 9: Thank you for pointing this out. We improve the English within the text.
Round 2
Reviewer 1 Report
Comments and Suggestions for Authors
The author did not provide any WGS data for ST confirmation; however, I strongly believe that WGS data is required for ST determination. The author only provided the prevalence male-female ratio, nothing else. As a result, the article's quality does not support publication in MICROORGANISM, the top journal in MDPI. The article should be rejected.
Author Response
Comment 1: The author did not provide any WGS data for ST confirmation; however, I strongly believe that WGS data is required for ST determination.
Respond 1: Thank you for pointing this out. We agree with this comment. We already submitted the raw WGS data to the NCBI SRA. The Bioproject ID for 2018 is PRJNA1217336 and for 2022 is PRJNA1217803.
Comment 2: The author only provided the prevalence male-female ratio, nothing else.
Respond 2: Thank you for bringing this to our attention. We respectfully disagree with the comment. The demographic information is only one component of our manuscript (Lines 135-153, 228-244). In other sections, we present: susceptibility trends (Lines 157-184, 290-339), including MDR/XDR cases (Lines 174-184, 332-336), basic molecular mapping (sequence types) (Lines 186-208, 345-368), a combination of phenotypic antimicrobial susceptibility testing with NG-MAST genotyping (Lines 204-208). These additional analyses provide a comprehensive overview of the subject matter, extending well beyond demographic data.
Reviewer 2 Report
Comments and Suggestions for Authors
The authors have adequately addressed all my comments. The manuscript has significantly improved. I have no further remarks and believe the paper can be accepted, pending the Editor's final decision.
Author Response
Comment 1: The authors have adequately addressed all my comments. The manuscript has significantly improved. I have no further remarks and believe the paper can be accepted, pending the Editor's final decision.
Respond 1: Thank you for your comment